

# Mechanical response microRNA-145a-5p alleviates osteoarthritis by inhibiting inflammation and promoting chondrogenesis

Jingke Du[1,2,3,*], Zhen Zhang[1,2,3,*], Danyang Bai[4], Ting Zhu[1,2,3] and Dong Jiang[1,2,3]

[1] Department of Sports Medicine, Peking University Third Hospital, Institute of Sports Medicine of Peking University, Beijing, China
[2] Beijing Key Laboratory of Sports Injuries, Beijing, China
[3] Engineering Research Center of Sports Trauma Treatment Technology and Devices, Ministry of Education, Beijing, China
[4] Department of Orthopedic Oncology, Shanghai Changzheng Hospital, Naval Military Medical University, Shanghai, China
* These authors contributed equally to this work.

Corresponding author
Dong Jiang, bysyjiangdong@126.com

## ABSTRACT

**Background:** Osteoarthritis is characterized by cartilage wear or absence and is usually initiated by inflammation and abnormal mechanical stimulation. MicroRNAs have been identified as the main regulators of osteoarthritis, but the influence of miR-145a-5p on osteoarthritis has not been elucidated. In this study, we focused on the role of miR-145a-5p in cartilage.

**Methods:** Destabilization of the medial meniscus surgery (DMM) and RNA fluorescence *in situ* hybridization (RNA FISH) were performed to detect the expression level of miR-145a-5p in osteoarthritis. Interleukin-1β was used to simulate the inflammatory environment *in vitro*. The Flexcell device was used for mechanical stimulation. Agomir-145a-5p was injected intra-articularly into the DMM-induced osteoarthritis mouse model. Histopathological examinations, and molecular biology techniques were used to investigate the underlying mechanisms.

**Results:** The expression of miR-145a-5p was decreased in osteoarthritis mice, whereas its expression increased with prolonged chondrogenesis. Then, studies *in vitro* also confirmed the pro-chondrogenesis and interleukin-1β inhibitory ability of miR-145a-5p. Additionally, miR-145a-5p can be regulated by cyclic stretch stress, with physiological mechanical stimulation promotes, but excessive mechanical stimulation suppresses its expression. In addition, miR-145a-5p rescues DMM-induced osteoarthritis progression, which was observed through the intra-articular injection of agomiR-145a-5p.

**Conclusions:** MiR-145a-5p, a mechanical responder, alleviates osteoarthritis progression through promoting chondrogenesis and alleviating inflammation response. And intra-articular injection of miR-145a-5p alleviates osteoarthritis progression. These findings suggest that miR-145a-5p is a promising target for the treatment of osteoarthritis.

# INTRODUCTION

Osteoarthritis (OA), characterized by pathological changes in all joint tissues (*Wood et al., 2023*; *Motta et al., 2023*), including cartilage, subchondral bone, and synovial membrane, has become the leading chronic orthopedic condition worldwide (*DeJulius et al., 2024*; *Messier et al., 2022*). As a complex disease, OA pathogenesis includes initial injury and frequent biomechanical damage to any of the joint tissues, which results in the release of cytokines and leads to the activation of different signaling pathways that damage cartilage (*Sanchez-Lopez et al., 2022*).

The inflammatory mediators that participate in the progression of OA mainly include interleukin-1β (IL-1β), interleukin-6 (IL-6), and tumor necrosis factor (TNF), which are highly expressed in OA synovial fluid (*Chou et al., 2020*; *Lv et al., 2019*). Research has confirmed that the secretion of IL-1β downregulates the synthesis of type II collagen (COL2) and aggrecan (ACAN), thereby inhibiting chondrogenesis (*Liu et al., 2023*; *He et al., 2022*). In addition, IL-1β is commonly used *in vitro* to mimic pro-inflammatory and pro-catabolic chondrocyte phenotypes (*Defois et al., 2023*). So, anti-inflammation is one of the most essential methods in OA treatment. As one of the critical anti-inflammation factors, microRNAs (miRNAs) have been reported to participate in the development and progression of OA (*Swingler et al., 2019*). For example, *Ji et al. (2021)* constructed a nanocarrier to target the miR-141/200c cluster in chondrocytes to attenuate osteoarthritis development. Additionally, miR-204 has been identified as an ameliorator of OA pain by inhibiting SP1-LRP1 signaling (*Lu et al., 2023*).

Furthermore, as another leading cause of OA, biomechanical stimulation cannot be ignored in its initiation (*Hodgkinson et al., 2022*). In healthy cartilage, the mechanical stimulation generated by movement is integral to maintaining the homeostatic balance of chondrocytes. However, abnormal mechanical activity is harmful to the health of cartilage; for example, joint non-use can lead to harmful cartilage atrophy (*Vincent & Wann, 2019*), and excessive mechanical stimulation is a risk factor for the pathogenesis and progression of OA (*Chang et al., 2019*). As the regulator of post-transcriptional gene expression (*Giordano et al., 2020*), several miRs have been identified to participate in the mechanical response, such as miR-21 and miR-325-3p (*Li et al., 2017*; *Huang et al., 2012*). Determining the mechanism of how mechanical stimulation regulates joint homeostasis will help develop new approaches to treating OA (*Gargano et al., 2022*).

MiR-145a-5p, encoded by the MIR145 gene located on chromosome 5: 149,430,646–149,430,733 forward strands, has emerged as a key regulator in various diseases (*Kadkhoda & Ghafouri-Fard, 2022*; *Li et al., 2023*). As a highly homologous form of miR-145, which has been confirmed to modulate TNF-α-mediated signaling and cartilage matrix degradation (*Guo et al., 2024*; *Hu et al., 2017*), miR-145a-5p has been shown to influence the pathogenesis of many inflammatory diseases, such as chronic obstructive pulmonary disease (COPD), aplastic anemia, and rheumatoid arthritis (*Kadkhoda & Ghafouri-Fard, 2022*). Considering the role of miR-145a-5p in inflammatory

diseases, it may be involved in the occurrence of OA and the mechanical response, which deserves further study.

In this study, several experiments were conducted to reveal the function of miR-145a-5p in cartilage both *in vivo* and *in vitro*, which aimed to illuminate the function of miR-145a-5p in OA development and cartilage hemostasis.

# MATERIALS AND METHODS

## Animal experiments

After inducing anesthesia *via* intraperitoneal injection of sodium pentobarbital (50 mg/kg body weight), followed by a stabilization period of ≥10 min to ensure surgical anesthesia, destabilization of the medial meniscus (DMM) or sham surgery was performed on forty 8-week-old male C57BL/6J mice (obtained from the department of laboratory animal science of Peking University Health Science Center) using standardized surgical protocols, as previously described (*Glasson, Blanchet & Morris, 2007*). The protocol was complied with the Guide for the Care and Use of Laboratory Animals published by the National Academy Press (National Institutes of Health Publication No. 85–23, revised 1996), and the procedures involving animal experiments were reviewed and approved by the Peking University Biomedical Ethics Committee (Grant number: PUIRB-LA2022629). The mice were placed separately (five mice per cage) and kept at ambient temperature (23 ± 3 °C) with 12-h light/dark cycle and free access to a standard pellet diet and water. For agomir delivery, eighteen mice in the DMM group were randomly assigned to experimental groups (DMM+NC-agomir, DMM+agomir-145a-5p, OA) using a computer-generated random number sequence, and intra-articular injection of 10 μg of agomir-NC (miR-NC, GenePharma, Suzhou, China), or agomir-145a-5p (B06023; miR-145a-5p, GenePharma, Suzhou, China) at 1-, 3-, 5-, and 7-weeks post-operation, the flowchart was shown in Fig. S1. Nine weeks after DMM surgery, the mice were anesthetized with pentobarbital sodium (50 mg/kg body weight) for at least 10 min, followed by humane euthanasia *via* cervical dislocation to minimize suffering, and then their knee joints were collected. No surviving animals remained at study conclusion.

## Histological, immunofluorescence, and RNA FISH analyses

Mouse knee joints were collected and fixed immediately with 4% paraformaldehyde for 48 h, decalcified with 10% EDTA for 10 days, dehydrated for paraffin embedding, and continuous sagittal cross-sections sequentially from medial to lateral orientations were collected. These sections were cut to a thickness of 5 microns, dewaxed, and stained with hematoxylin and eosin (HE), toluidine blue, safranin O and fast green. HE and safranin O and fast green stained tissue sections were used for OARSI scoring. The OARSI scoring system 50 was performed by two observers blinded to the experimental groups to evaluate cartilage destruction in the medial joints (*Arden et al., 2021*). Immunofluorescence (IF) staining was performed on paraffin-embedded joint sections or cultured cells. After antigen retrieval (for tissues) or permeabilization (for cells), samples were blocked with 5% BSA and 10% normal goat serum, followed by incubation with primary antibodies against YAP (1:50, sc-101199), MMP13 (1:50, 18165-1-AP), and COL2A1 (1:100, ARG20787)

overnight at 4 °C. Species-specific Alexa Fluor-conjugated secondary antibodies (1:500, 4408S/8889; Cell Signaling Technology, Danvers, MA, USA) were applied for 1 h at room temperature, and nuclei were counterstained with DAPI (C0065; Solarbio, Beijing, China). The RNA FISH assay was performed *via* an RNA FISH kit (GenePharma, Suzhou, China) according to the manufacturer's instructions. Images were acquired on a ZEISS LSM 880 with Airyscan (Carl Zeiss Microscopy GmbH, Jena, Germany). Fluorescence intensity was measured in 3–5 microscopic fields per joint using ImageJ software, with all analyses conducted by an operator blinded to the group identity. The median value for each mouse was then calculated and reported (*Zhang et al., 2022*).

## Cell culture and treatment

ATDC5 cells (American Type Culture Collection, ATCC, Manassas, VA, USA) were maintained at 37 °C in a humidified incubator with 5% $CO_2$ and were grown in Dulbecco's modified Eagle's medium (DMEM, Gibco, Waltham, MA, USA) supplemented with 10% fetal bovine serum (FBS, SV30208.02; HyClone, Logan, UT, USA) before differentiation. To induce chondrogenic differeclenium (NO. 41400045; Thermo Fisher, Waltham, MA, USA), was added to the growth medium of ATDC5 cells, and alcian blue staining and q-PCR were performed after 7 days. For the treatment of IL-1β, 10 ng/ml IL-1β (MCE, HY-P7073A) was added to the chondrogenic differentiation induction medium of ADTC5. The medium was changed every 2 days (*Shen et al., 2017*).

Primary human articular chondrocytes (HACs) were isolated from macroscopically non-lesional cartilage fragments obtained during arthroscopic procedures, following written informed consent under protocols approved by the Peking University Third Hospital Medicine Science Research Ethics Committee (grant number: M2023779). Tissues were processed within 4 h. After thorough washing and dissection to isolate the chondral layer, fragments were minced and digested with 0.2% collagenase type II (17101015; Gibco, Waltham, MA, USA) in DMEM overnight (37 °C, 5% $CO_2$). The resulting suspension was filtered (100 μm), washed extensively in PBS, and pelleted cells were resuspended in DMEM supplemented with 15% fetal bovine serum (SV30208.02; FBS, HyClone, Logan, Utah, USA) and 1% penicillin-streptomycin (C0222; Beyotime, Shanghai, China). Cells were cultured at 37 °C, 5% $CO_2$ and used at passage 1 (P1) or P2 for experiments to minimize dedifferentiation (*Gan et al., 2024*).

## Cyclic stretch stress

ATDC5 cells in three groups (Ctrl, 10% Stretch and 20% Stretch) were cultured in BioFlex 6-well culture plates precoated with type I collagen for 24 h. When the ATDC5 cells reached a density of 50–60%, they were stretched with a Flexcell-4000T vacuum stretching device at an amplitude of 10% (normal mechanical stimulation, 10% stretch) or 20% (*Huang et al., 2025*) (excessive mechanical stimulation, 20% stretch) elongation and a frequency of 0.5 Hz for 4 h per day for 4 days (*Na et al., 2024*), ctrl group keep static in the same incubator.

## Transfection of miR-145a-5p mimics, inhibitors, and negative controls

When ATDC5 cells in 12-well plates reached 30–40% confluence, 50 nM mmu-miR-145a-5p mimics (miR-mimic), inhibitors (miR-inhibitor), or negative controls (miR-NC) (all purchased from GenePharma) were transfected into ATDC5 cells *via* Lipofectamine 3000 (L3000008; Invitrogen, Carlsbad, CA, USA) according to the manufacturer's instructions, after which the medium was changed to chondrogenic differentiation induction medium after 6 h.

## RNA extraction and quantitative real-time PCR

Total RNA was extracted *via* TRIzol reagent (15596026CN; Invitrogen, Carlsbad, CA, USA) according to the manufacturer's instructions. The RNA concentration and purity were assessed *via* a Nanodrop spectrophotometer (ND-1000; Thermo Scientific, Waltham, MA, USA). The values of A260/A280 ratio between 1.8 and 2.0 indicating high purity. Reverse transcription reactions were performed using two distinct protocols for mRNA and miRNA synthesis. For mRNA cDNA synthesis, total RNA samples (2 µg) were reverse transcribed with GoScript$^{TM}$ Reverse Transcriptase (A2800; Promega, Madison, WI, USA) following the manufacturer's protocol. The reaction system (20 µL final volume) contained 2 µg of purified total RNA samples, 4 µL GoScript$^{TM}$ Reaction Buffer (Random Primer), 2 µL GoScript$^{TM}$ Enzyme Mix, and nuclease-free water. Thermal cycling conditions were programmed as follows: primer annealing at 25 °C for 5 min, cDNA synthesis at 42 °C for 60 min, and enzyme inactivation at 70 °C for 15 min. For miRNA-specific cDNA synthesis, the miRNA First Strand cDNA Synthesis Kit (B532451; Sangon Biotech, Shanghai, China) was employed. A 20 µL reaction mixture containing 2 µg total RNA, 10 µL 2× miRNA P-RT Solution Mix, and 2 µL miRNA P-RT Enzyme Mix was subjected to the following thermal profile: 37 °C for 60 min followed by enzyme denaturation at 85 °C for 5 min. All cDNA products were stored at −20 °C for subsequent analysis. Quantitative real-time PCR was performed using an ABI 7500 system (Applied Biosystems, Foster City, CA, USA) with NovoStart® SYBR Green SuperMix Plus (E166-01B; Novoprotein, Suzhou, China). Each 20 µL reaction contained 2 µL cDNA template, 10 µL SYBR Green Master Mix, 1 µL (200 nM) forward/reverse primers (Sangon Biotech, Shanghai, China sequences in Table S1), and 7 µL nuclease-free water. The thermal protocol consisted of an initial denaturation at 95 °C for 1 min, followed by 40 cycles of 95 °C for 10 s and 60 °C for 30 s. The expression values were normalized to those of Gapdh or U6 *via* the $2^{-\Delta\Delta Ct}$ method (*Kan et al., 2022*).

## Dual-luciferase assay

ATDC5 cells were cultured at a density of $1 \times 10^5$ cells/well in 12-well culture plates and transfected with 2 µg of dual-luciferase reporter pmirGLO-IL6-WT or pmirGLO-IL6-Mut (all purchased from D-Nano Therapeutics), and co-transfected with 500 nM mmu-miR-145a-5p mimics or miR-NC using Lipofectamine 3000 (L3000008; Invitrogen, Carlsbad, CA, USA) according to the manufacturer's protocol. Six hours post-transfection, the

transfection medium was removed and replenished with growth medium. Forty-eight hours post-transfection, luciferase activity was measured using the Dual Luciferase Reporter Assay Kit (DL101-01; Vazyme, Nanjing, China). Firefly and renilla luciferase activity were detected by a microplate reader (Varioskan Flash, Thermo, Waltham, MA, USA) Firefly luciferase activity was normalized to renilla luciferase activity (*Jiang et al., 2022*).

## Statistical analysis

All data were obtained from three independent biological replicates, each with technical triplicates. The data are presented as the means ± standard deviations (SDs). One-way ANOVA with Tukey's *post-hoc* test was applied for multi-group comparisons when data met normality (Shapiro-Wilk test) and homogeneity of variance (Levene's test) assumptions. Student's t test was used to analyze the differences between two groups. Statistical analyses were performed *via* GraphPad Prism 9 software. $p < 0.05$ was considered statistically significant.

## RESULTS

### MiR-145a-5p is decreased in osteoarthritis

To investigate the function of miR-145a-5p, we design a series of experiments to confirm its role in OA progression. The flow diagram is shown in Fig. 1A. Toluidine blue staining reveals thinner and rougher articular cartilage in the OA group (Fig. 1B), confirming the successful establishment of the osteoarthritis mouse model. Then, qRT-PCR is performed to detect the expression level of miR-145a-5p *in vivo*; Fig. 1C shows that OA cartilage has a lower miR-145a-5p level. As depicted in Fig. 1D, the fluorescence intensity of miR-145a-5p in the cartilage of OA mice is faint, which is consistent with the statistical results presented in Fig. 1E.

### MiR-145a-5p promotes chondrogenic differentiation and regulates the expression of *Nrf2* and *Il-6*

MiR-145a-5p expression levels were detected at different time points of chondrogenesis (Fig. 2A). With increasing differentiation time, the miR-145a-5p expression levels gradually increased. Subsequently, miR-145a-5p mimic- and inhibitor-transfected cells were used to detect the effect of miR-145a-5p on chondrogenesis. As shown in Fig. 2C, after transfection with the miR-145a-5p mimic, the miR-145a-5p expression level significantly increased, whereas the miR-145a-5p expression level decreased in the inhibitor group. qRT–PCR was performed 3days after cell chondrogenic induction, as shown in Figs. 2D–2F. miR-145a-5p mimic promotes the expression of *Sox9, Acan*, and *Col2*. Moreover, alcian blue staining reveals greater chondrogenic differentiation ability in the miR-145a-5p-mimic group than in the control group (Fig. 2B). The downstream genes of miR-145a-5p were predicted *via* miRWalk (http://mirwalk.umm.uni-heidelberg.de), and the predicted binding sequences were shown in Fig. 2G. As shown in Figs. 2H, 2I, the expression levels of *Nrf2* and *Il-6* were increased in the miR-145a-5p-inhibitor group but decreased in the mimic group, indicating that miR-145a-5p might alleviate OA through

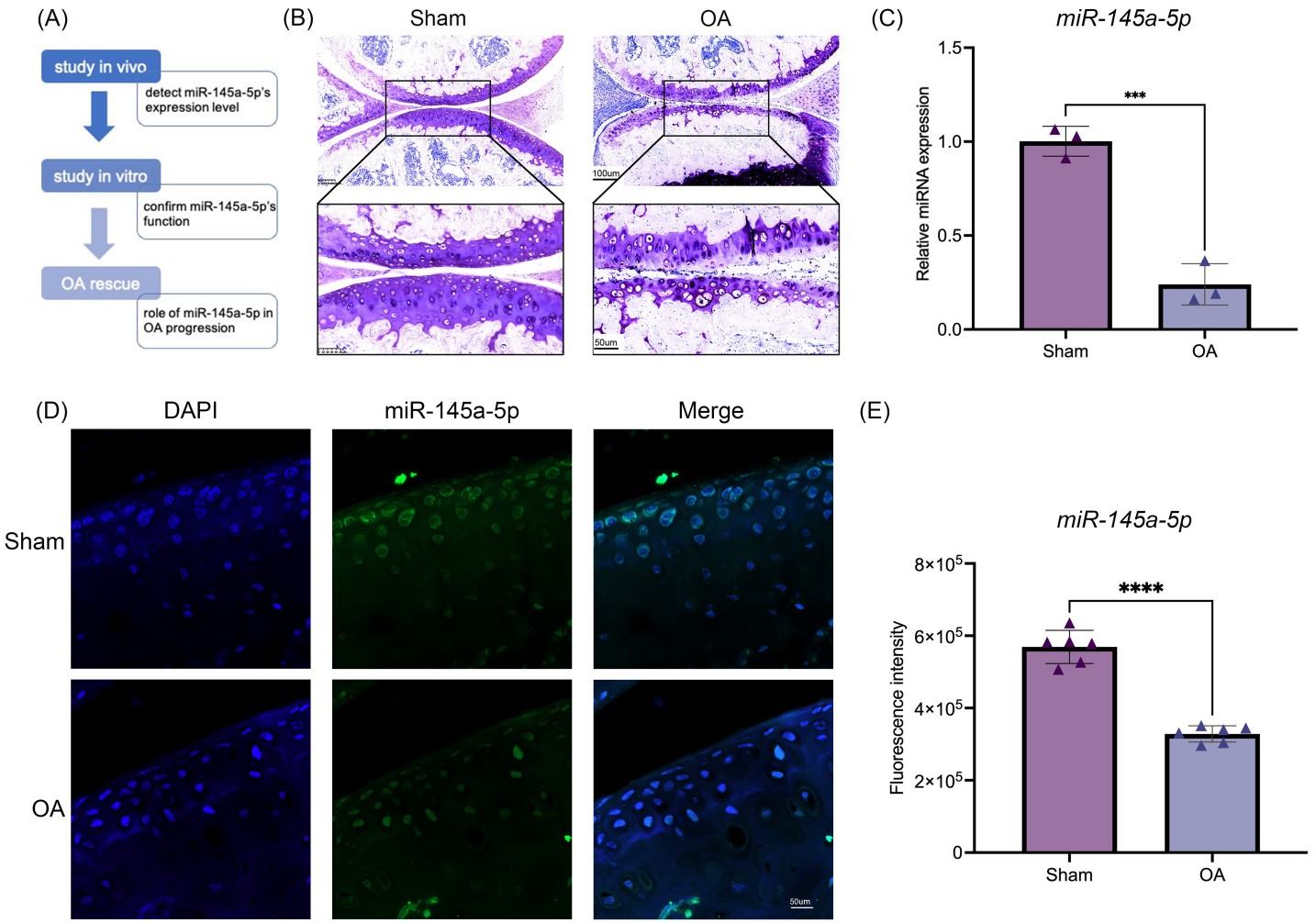

**Figure 1** **MiR-145a-5p expression is decreased in osteoarthritis.** (A) The study design flow diagram. (B) Toluidine blue staining of the sham and OA groups. (C) qRT-PCR results of cartilage miR-145a-5p expression levels. (D) MiR-145a-5p expression in the sham and OA groups was detected *via* RNA FISH. (E) The fluorescence intensity in the sham and OA groups was calculated using ImageJ 2.3.0. (***$p < 0.005$ and ****$p < 0.001$).

*Nrf2* and *Il-6*. To validate the predicted miR-145a-5p binding site in the IL-6 3'UTR, we conducted dual-luciferase reporter assays. Co-transfection of miR-145a-5p mimic with the wild-type IL-6 3'UTR reporter significantly suppressed luciferase activity, whereas the mutant reporter showed no response, demonstrating direct interaction between miR-145a-5p and the IL-6 3'UTR (Fig. S2).

## MiR-145a-5p can partially reverse the chondrogenic inhibition caused by IL-1β

IL-1β was added to the culture medium of ATDC5 cells. Results showed that IL-1β suppresses the expression of miR-145a-5p (Figs. 3A, 3B). Concurrently, the expression levels of *Sox9, Col2*, and *Acan* also indicate the inhibitory effect of IL-1β on chondrogenesis

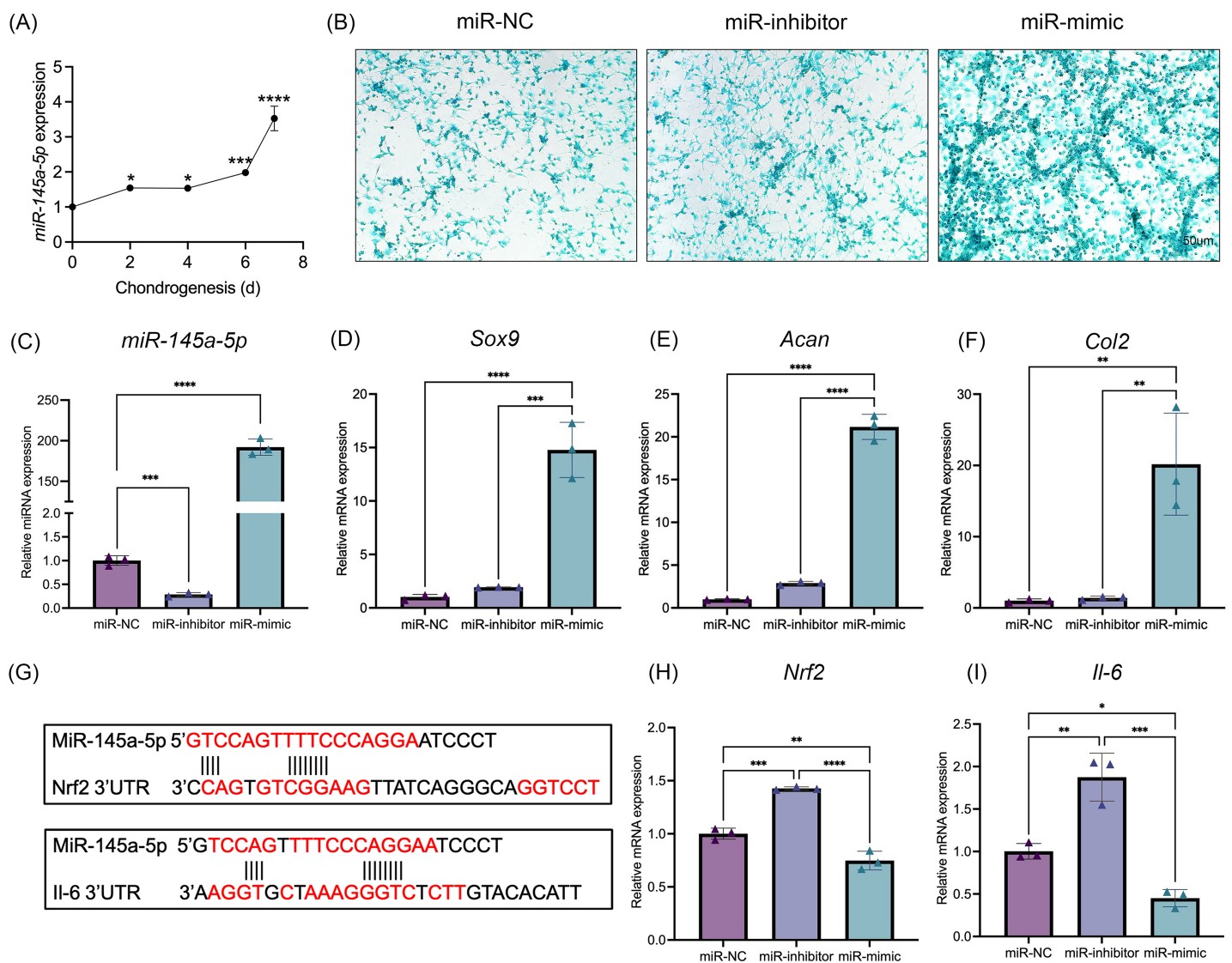

**Figure 2** **MiR-145a-5p promotes chondrogenic differentiation and regulates the expression of Nrf2 and Il-6.** (A) The miR-145a-5p expression level was detected *via* qRT−PCR during chondrogenesis. (B) Alcian blue staining was performed in the miR-NC, miR-inhibitor, and miR-mimic groups. (C–F) The expression of *miR-145a-5p, Sox9, Acan*, and *Col2* was detected after transfection with the miR-145a-5p negative control (miR-NC), miR-145a-5p inhibitor (miR-inhibitor) or miR-145a-5p mimic (miR-mimic). (G) The binding sequences of Nrf2 and Il-6 with miR-145a-5p were predicted. (H-I) *Nrf2* and *Il-6* levels were detected in the three groups. (*$p < 0.05$, **$p < 0.01$, ***$p < 0.005$ and ****$p < 0.001$).

(Fig. 3C). Consistent with these findings, IL-1β similarly suppressed *miR-145a-5p, Sox9, Col2, and Acan* expression in HACs (Fig. S3). As shown in Fig. 3D, IL-1β reduced the expression of *Sox9* and *Col2*, whereas miR-145a-5p increased the expression of *Sox9, Col2*, and *Acan*. Moreover, miR-145a-5p alleviated the inhibitory effect of IL-1β on chondrogenesis. Subsequently, alcian blue staining was performed on the cells. As shown in Fig. 3E, miR-145a-5p partially reversed the inhibitory effect of IL-1β on cartilage differentiation.

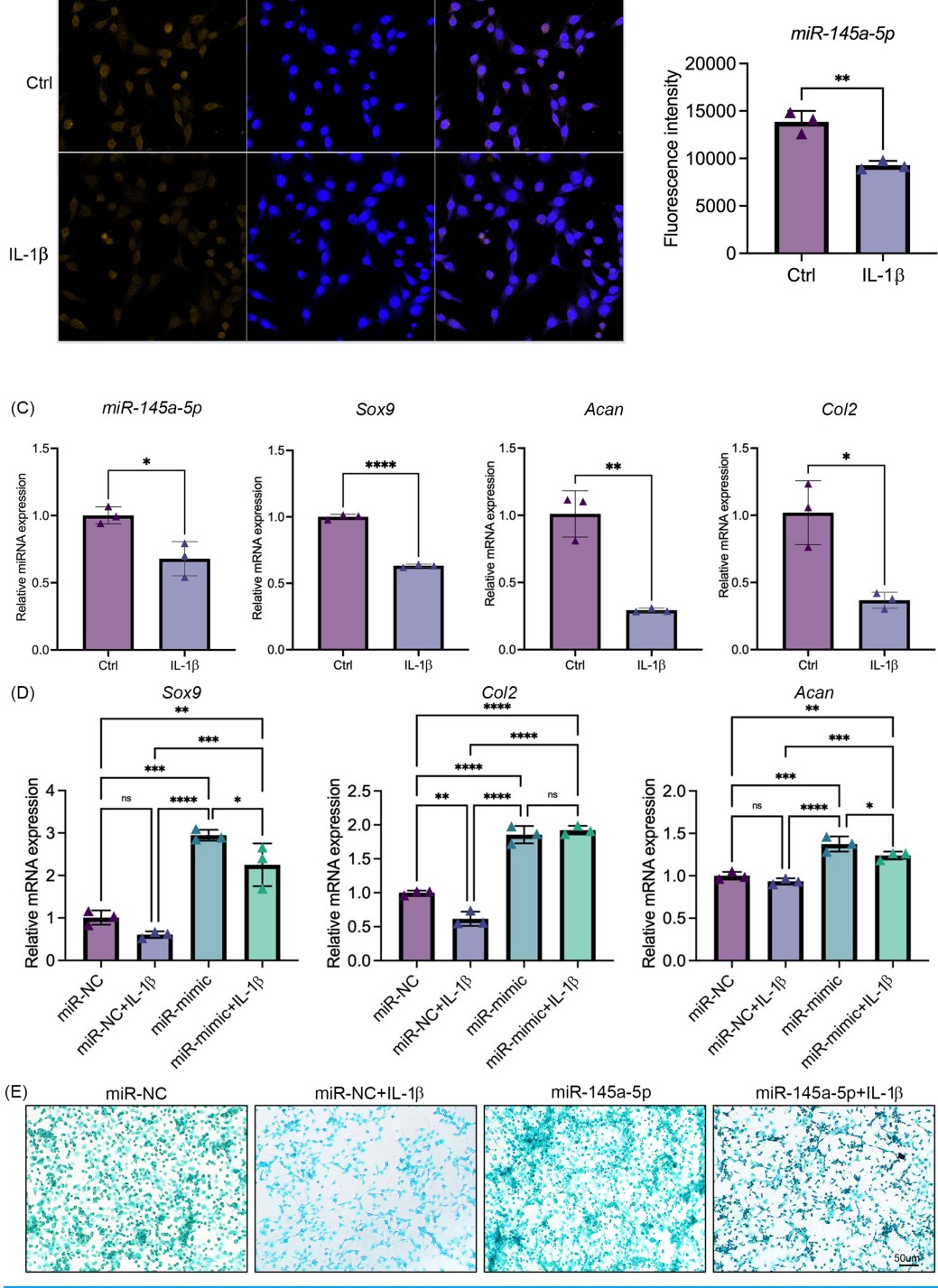

**Figure 3 MiR-145a-5p can partially reverse the chondrogenic inhibition caused by IL-1 b.** (A) MiR-145a-5p in ATDC5 cells was detected *via* RNA FISH after treatment with IL-1 β. (B) Fluorescence intensity was calculated using ImageJ. (C) The expression levels of *miR-145a-5p, Sox9, Acan*, and *Col2* in the Ctrl and IL-1 β groups were detected *via* qRT-PCR. (D) The expression levels of *Sox9, Col2* and *Acan* in the miR-NC and miR-mimic groups with or without IL-1 β were detected. (E) Alcian blue staining was performed in the four groups. (*$p < 0.05$, **$p < 0.01$, ***$p < 0.005$ and ****$p < 0.001$).

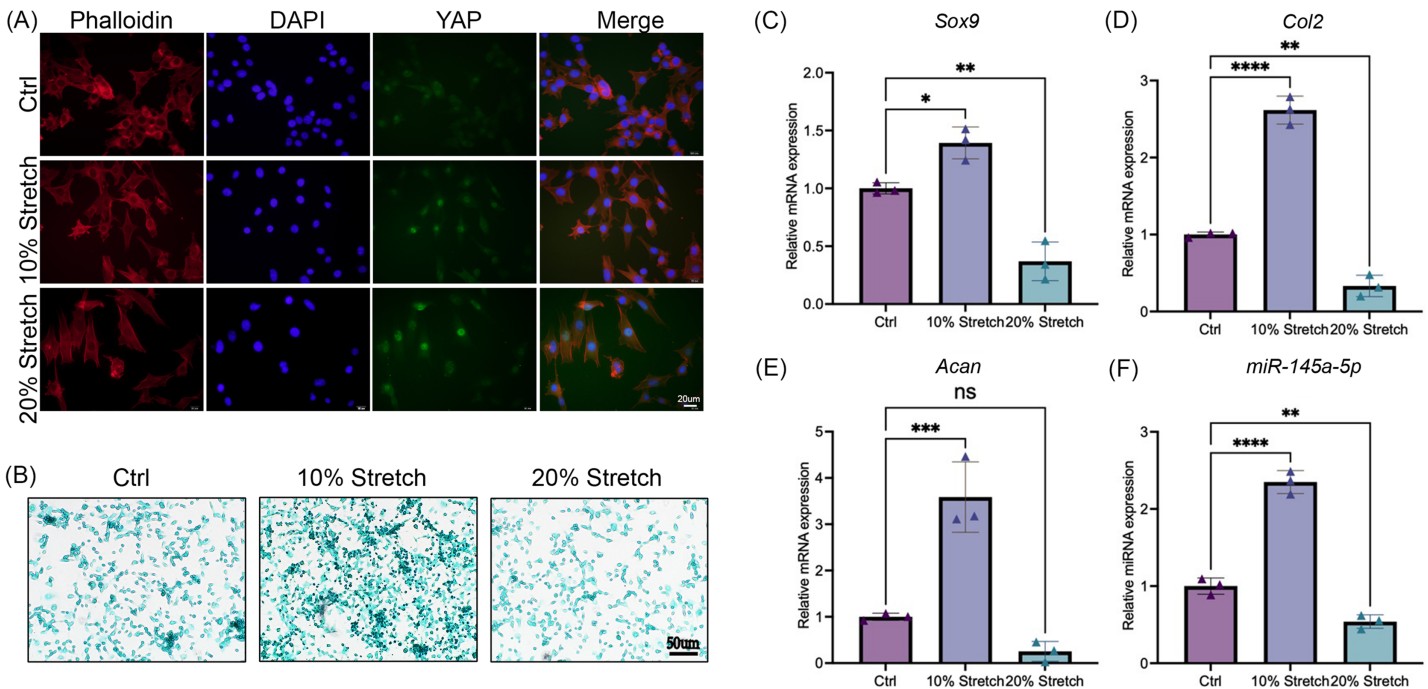

**Figure 4 Cyclic stretch stress regulates chondrogenesis and the expression of miR-145a-5p.** (A) Immunofluorescence of YAP in three groups after stimulation with tensile mechanics. (B) Alcian blue staining was performed in the Ctrl and Stretch groups. (C–F) *Sox9, Acan, Col2*, and *miR-145a-5p* expression in ATDC5 cells was detected after cyclic stretch stress. (*$p < 0.05$, **$p < 0.01$, ***$p < 0.005$ and ****$p < 0.001$).

## Cyclic stretch stress regulates chondrogenesis and the expression of miR-145a-5p

After stimulation with tensile mechanics, the intracellular distribution of YAP was detected by immunofluorescence to confirm the mechanical response of ATDC5 cells (*Dupont et al., 2011*). As shown in Fig. 4A, with the increase of tensile mechanics, the distribution of YAP in the nucleus was enhanced, and the 20% stretching significantly promoted the nucleolar residence of YAP, which gives us a clue that ATDC5 can receive the mechanical stimulation. Subsequently, alcian blue staining revealed that the cells in the 10% stretch group exhibited greater chondrogenic ability (Fig. 4B), while the 20% stretch inhibits chondrogenesis. As shown in Figs. 4C–4F, the application of physiological tensile mechanics (10% stretch) promoted the expression of *miR-145a-5p*, *Sox9*, *Acan*, and *Col2*, whereas overload (20% stretch) decreased their expression.

## MiR-145a-5p rescues OA progression *in vivo*

To verify the role of miR-145a-5p in OA progression, ago-miR-145a-5p or ago-miR-NC was intra-articularly injected into the knees of mice after DMM surgery. As shown in Figs. 5A, 5B, the expression level of miR-145a-5p in the cartilage was increased in the miR-145a-5p group, and the administration of ago-miR-145a-5p indeed alleviated cartilage destruction and decreased OARSI scores at 9 weeks post DMM surgery (Figs. 5C, 5D).

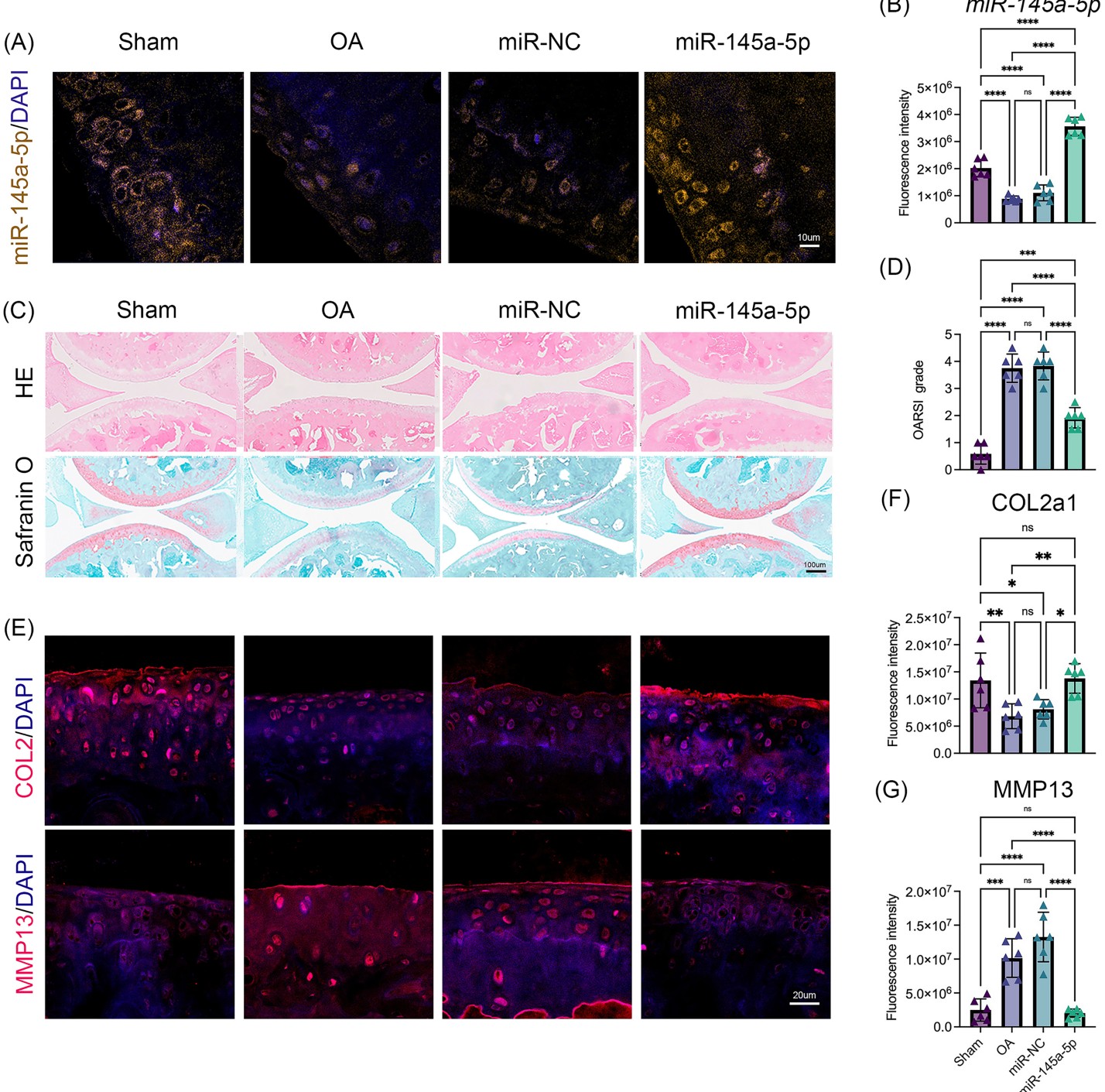

**Figure 5 MiR-145a-5p rescues OA progression *in vivo*.** (A) RNA FISH was performed to detect the expression level of miR-145a-5p in the four groups. (B) The fluorescence intensity of miR-145a-5p was calculated *via* ImageJ. (C and D) Representative images of HE and safranin O/fast green staining of joint sections and OARSI scores. (E–G) Representative images of IHC staining of joint sections and the fluorescence intensity results of COL2 and MMP13. (*$p < 0.05$, **$p < 0.01$, ***$p < 0.005$ and ****$p < 0.001$).

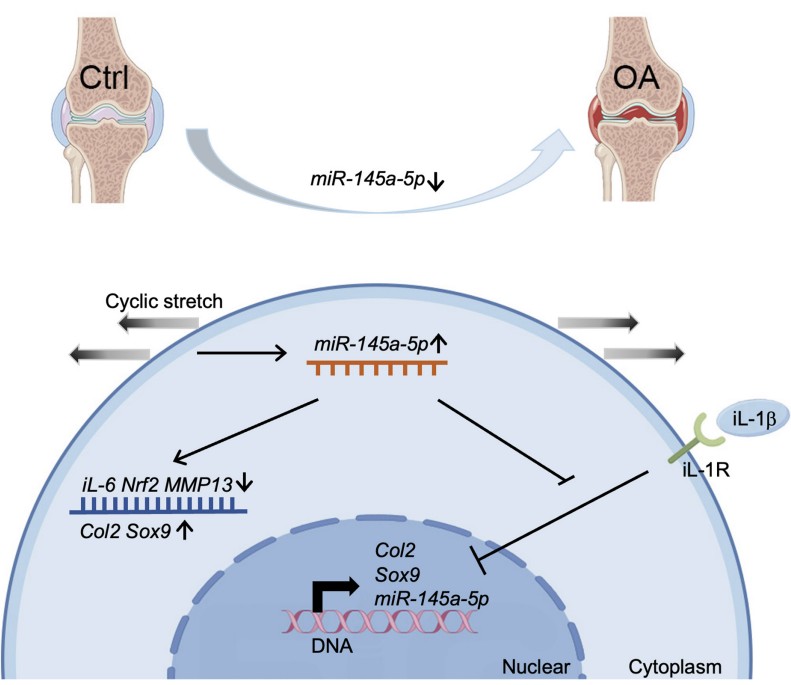

**Figure 6 Schematic diagram representing the mechanism by which miR-145a-5p alleviates OA development.** MiR-145a-5p expression is downregulated in OA chondrocytes, it is also regulated by cyclic stretch stress and can impair IL-1 β's function. Intra-articular injection of miR-145a-5p alleviates OA progression.

Moreover, miR-145a-5p partially improved the expression of COL2 and suppressed the expression of MMP13 (Figs. 5E–5G).

## DISCUSSION

Our study demonstrates a significant decrease in miR-145a-5p expression in OA cartilage, contrasting with its upregulation during chondrogenesis. We further establish that miR-145a-5p can partially mitigate the detrimental effects of IL-1β on chondrogenesis. Crucially, we identify cyclic stretch stress as a key regulator of miR-145a-5p expression, with 10% stretch promoting and 20% stretch suppressing its levels. *In vivo* studies confirm the OA-alleviating function of miR-145a-5p (Fig. 6), highlighting its potential as a novel therapeutic target.

Our results demonstrate that miR-145a-5p effectively attenuates IL-1β-induced cartilage degradation in ATDC5 cells, consistent with prior findings showing the expression of miR-145 increased in OA chondrocytes and responded to IL-1β stimulation (*Yang et al., 2014*). Similar to miR-365 and miR-144-3p (*Lin et al., 2021*; *Hwang et al., 2017*), miR-145a-5p exhibits reduced expression in OA cartilage. While miR-365 inhibits IL-1β-induced HIF-2α upregulation and targets HDAC4 (*Chen & Wu, 2019*), and miR-144-3p downregulates IL-1β *via* MAPK, PI3K/Akt, and NF-κB pathways, we observed a comparable negative regulatory relationship between IL-1β and miR-145a-5p. The anti-inflammatory effects of miR-145a-5p appear mediated through the direct regulation

of IL-6. Furthermore, this study reveals the novel mechanoresponsive nature of miR-145a-5p. Reported targets of miR-145a-5p include the NF-κB signaling pathway (regulating pyroptosis) (*Yao et al., 2022*) and SMAD5 (promoting postinfarction revascularization) (*Long et al., 2022*), suggesting potential involvement of NF-κB signaling in miR-145a-5p's actions during chondrogenesis or OA progression (*Ni et al., 2023*). Other miRNAs including miR-29a (*Li et al., 2016*) and miR-448 (*Yang et al., 2018*) also exhibit chondroprotective properties, whereas miR-217 (*Papageorgiou et al., 2023*) and miR-320a (*Jin et al., 2017*) promote OA progression through SIRT1 and PBX3 regulation respectively, both being upregulated in OA.

Consisting with studies that reported anti-inflammatory properties of miR-145a-5p, our results indicate it counteracts the inhibitory effect of IL-1β on chondrogenesis (*Yao et al., 2022*; *Ramelli et al., 2020*). IL-1β, a pivotal inflammatory factor in OA cartilage and synovial (*Sanchez-Lopez et al., 2022*), exacerbates OA by inducing oxidative stress and activating the MAPK, NF-κB, and Wnt pathways (*Zhu et al., 2020*; *Feng et al., 2017*). This activation leads to the upregulation of IL-6 (*Zhu et al., 2020*), MMP3, MMP9, and NRF2 (*Wu et al., 2018*; *Zhang et al., 2020*). Our study demonstrates that miR-145a-5p attenuates IL-1β's function and suppresses the expression of *Il-6* and *Nrf2* expression (Figs. 2H, 2I), suggesting its potential involvement in modulating oxidative stress, MAPK, NF-κB, and Wnt pathways (*Lane & Felson, 2020*; *Arra et al., 2020*). Using IL-1β-treated ATDC5 cells to model inflammation *in vitro*, we confirmed miR-145a-5p's anti-inflammatory effect. Notably, the miR-145a-5p mimic prevented IL-1β-induced suppression of Col2 expression (Fig. 3D), indicating its role in protecting cartilage matrix and promoting regeneration.

Applying cyclic stretch stress to ATDC5 cells revealed that 10% elongation promoted both chondrogenesis and miR-145a-5p expression, while 20% elongation exerted opposing effects. These results demonstrate that miR-145a-5p is a mechanical response factor that can be targeted to attenuate the mechanically induced pathological transformation of chondrocytes (*Sun et al., 2017*; *Pathak et al., 2014*). Several miRNAs, such as miR-335-5p (*Xie et al., 2025*) and miR-3085-3p (*Lai et al., 2025*) (upregulated by overload, accelerating OA), and miR-143-3p (*Yan et al., 2024*) (mechanically suppressed, inhibiting MSC chondrogenesis), have been implicated in mechanotransduction during OA. Our findings demonstrate that optimal mechanical stress enhances both miR-145a-5p expression and chondrogenic differentiation. However, the precise mechanisms governing mechanical regulation of miR-145a-5p expression *in vivo* remain unresolved, complicated by altered joint load distribution due to cartilage degeneration. These pathological changes expose chondrocytes to abnormal mechanical stimuli even under normal loading conditions (*Vincent, 2013*; *Guilak et al., 2018*). PIEZO1, a critical mechanosensor upregulated in chondrocytes during OA progression (*Hodgkinson et al., 2022*; *Lee et al., 2021*; *He et al., 2024*), is a likely mediator of miR-145a-5p expression, and such dysregulated mechano-signaling contributes to OA progression *via* chondrocyte phenotypic changes (*Grad et al., 2011*).

Intra-articular injection of miR-145a-5p into the DMM mouse knee alleviated OA progression and inhibited MMP13 expression. This aligns with reports that miR-145 attenuates TNF-α-driven cartilage degradation and suppresses stroma-degrading enzymes

(MMP-3, MMP-13, Adamts-5) by targeting MKK4 (*Hu et al., 2017*). These findings position miR-145a-5p as a promising therapeutic target for OA. MicroRNAs have demonstrated therapeutic potential *in vivo* for various diseases (*Gargano et al., 2021*, *2023*). *Endisha et al. (2021)* reported that the level of miR-34a-5p was significantly increased in the plasma, cartilage, and synovium of patients with late-stage OA. That intra-articular miR-34a-5p antisense oligonucleotide had cartilage-protective effects on DMM and high-fat diet/DMM models (*Endisha et al., 2021*). MiR-21 has been identified as a critical regulator of cancer and inflammation (*Olivieri et al., 2021*; *Yu et al., 2015*). All of these give a clue that miRNAs can be a powerful method in treating disease (*Oliviero et al., 2019*). Moreover, several studies have developed new techniques for delivering miRs to improve available therapies (*Modica et al., 2021*; *Tan et al., 2021*). The limited effect of miR-145a-5p *in vivo* may be partly due to insufficient targeted delivery, and more effective methods need to be developed to improve its cartilage-targeting ability, which is a promising method for OA treatment.

To our knowledge, this is the first study elucidating the role of miR-145a-5p in cartilage metabolism and mechanical response. However, there are several limitations in this study. First, we only detected the effect of miR-145a-5p on chondrogenesis. It may also play a role in cell apoptosis, autophagy, and other phenotypes, so further studies are needed. Second, our study examined the mechanoresponsive role of miR-145a-5p only *in vitro*, while the mechanical parameters within joint cartilage during disease progression remain uncharacterized. Advanced imaging techniques or implantable sensors may help elucidate the dynamic mechanical microenvironment that regulates miR-145a-5p activity *in vivo*. Finally, the sample size ($n = 6$ per group) for some histological analyses may be considered modest. Future studies should further elucidate the mechanisms by which miR-145a-5p alleviates OA and validate these findings *in vivo* using larger cohorts to enhance reliability.

## CONCLUSIONS

In conclusion, the occurrence of osteoarthritis is related to a decrease in miR-145a-5p, and intra-articular injection of miR-145a-5p alleviates OA progression. Specifically, increased miR-145a-5p may slow the progression of osteoarthritis by inhibiting the function of inflammatory factors, such as IL-1β. MiR-145a-5p can also be considered a mechanical responder. Appropriate mechanical stimulation promotes but overload suppresses miR-145a-5p expression, which makes it a new potential target for OA treatment.

## LIST OF ABBREVIATIONS

| | |
|---|---|
| **OA** | Osteoarthritis |
| **RNA FISH** | RNA Fluorescence *in Situ* Hybridization |
| **NSAIDs** | nonsteroidal anti-inflammatory drugs |
| **IL-1β** | interleukin-1β |
| **IL-6** | interleukin-6 |
| **TNF** | tumor necrosis factor |
| **COL2** | type II collagen |
| **ACAN** | aggrecan |

| | |
|---|---|
| **miRNAs** | microRNAs |
| **COPD** | chronic obstructive pulmonary disease |
| **NASH** | nonalcoholic steatohepatitis |
| **DMM** | destabilization of the medial meniscus |
| **miR-NC** | agomir-NC |
| **H&E** | hematoxylin and eosin |
| **IF** | Immunofluorescence |
| **HACs** | Primary human articular chondrocytes |

## ACKNOWLEDGEMENTS

We thank our colleagues in the Beijing Key Laboratory of Sports Injuries for their help. We thank the Medical Research Center of Peking University Third Hospital for its technical support.

### Funding

This work is supported by the National Natural Science Foundation of China (82072428, 82472428), the National Key R&D Program of China (No. 2019YFB1706905) and the Natural Science Foundation of Beijing, China (7212132). The funders had no role in study design, data collection and analysis, decision to publish, or preparation of the manuscript.

### Grant Disclosures

The following grant information was disclosed by the authors:
National Natural Science Foundation of China: 82072428, 82472428.
National Key R&D Program of China: 2019YFB1706905.
Natural Science Foundation of Beijing, China: 7212132.

### Competing Interests

The authors declare that they have no competing interests.

### Author Contributions

- Jingke Du conceived and designed the experiments, performed the experiments, prepared figures and/or tables, authored or reviewed drafts of the article, and approved the final draft.
- Zhen Zhang conceived and designed the experiments, performed the experiments, authored or reviewed drafts of the article, and approved the final draft.
- Danyang Bai conceived and designed the experiments, analyzed the data, authored or reviewed drafts of the article, and approved the final draft.
- Ting Zhu performed the experiments, analyzed the data, authored or reviewed drafts of the article, and approved the final draft.
- Dong Jiang conceived and designed the experiments, authored or reviewed drafts of the article, and approved the final draft.

## Human Ethics

The following information was supplied relating to ethical approvals (*i.e.*, approving body and any reference numbers):

The primary human articular chondrocytes isolation procedures were reviewed and approved by the Peking University Third Hospital Medicine Science Research Ethics Committee (Grant number: M2023779).

## Animal Ethics

The following information was supplied relating to ethical approvals (*i.e.*, approving body and any reference numbers):

The animal experiment procedures were reviewed and approved by the Peking University Biomedical Ethics Committee (Grant number: PUIRB-LA2022629).

## Data Availability

Raw data is available in the Supplemental Files.

## Supplemental Information

Supplemental information for this article can be found online at http://dx.doi.org/10.7717/peerj.19905#supplemental-information.

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
