# Peer review of "Mechanical response microRNA-145a-5p alleviates osteoarthritis by inhibiting inflammation and promoting chondrogenesis"

_PeerJ, doi:10.7717/peerj.19905_

## Round 0.1 · original submission · Major Revisions

Dear Dr Du, kindly address the concerns raised by the reviewers and resubmit your manuscript.

Reviewer 1 ·

Basic reporting

The manuscript presents a well-structured study on the role of miR-145a-5p in osteoarthritis (OA), adhering to standard scientific reporting norms. The English language is fair to suggest editing. The introduction effectively contextualizes the research within the framework of OA pathogenesis, highlighting the gap in knowledge regarding miR-145a-5p. However, the literature review could be more concise, and some citations are too old. Figures and tables are relevant and of acceptable quality, with representative staining images including toluidine blue, safranin O/fast green, as well as quantitative data. However, some issues should be noted before acceptance.

Experimental design

The study design addresses the research question systematically, combining in vivo and in vitro approaches to explore miR-145a-5p function. Ethical approval for animal experiments is appropriately documented (PUIRB-LA2022629), and the use of standardized surgical protocols ensures reproducibility. However, several design considerations require attention:

(1) It is suggested to monitor the mechano-change in vivo.

(2) A concise flow chart is suggested to show the schedule plan of the animal study.

Validity of the findings

The results demonstrate consistent downregulation of miR-145a-5p in OA cartilage and its functional role in promoting chondrogenesis and inhibiting inflammation. The mechanistic link to mechanical stress is novel, showing that physiological vs. excessive stretch regulates miR-145a-5p expression. Intra-articular injection of agomir-145a-5p in DMM mice supports translational relevance by alleviating cartilage degradation. However, limitations in data robustness include:

(1) According to the literature, IL-6 is a pro-inflammatory factor, whereas Nrf2 is an anti-inflammatory one. Thus, the authors should double-check the results in Figure 2 H&I.

·

Basic reporting

-

Experimental design

-

Validity of the findings

-

Additional comments

This study reveals the dual roles of miR-145a-5p in cartilage metabolism and mechanical response, complementing the gap in the mechano-molecular regulatory network in the pathogenesis of OA. However, there are a few points that need to be further stated by the authors.

1. Although Mir-145a-5p has been proposed to function through targets such as Nrf2, IL-6, etc., direct experimental evidence (such as dual-luciferase reporter to verify the binding site) is lacking, and further studies are needed, and the specific signaling pathways (such as NF-κb, MAPK, etc.) are not clear. Further mechanism research is needed to enhance the reliability of the conclusion.

2. Only Atdc5 cells (a mouse chondrocyte cell line) were used, lacking primary human chondrocyte validation.

3. Does the “Overstretch” (20% strain) used in the experiment correspond to clinically relevant mechanical injuries (e.g., hypermobility, joint instability)? Interpretation of the correlation between mechanical stimulation and OA pathology.

4. Some in vivo experiments (e.g., OARSI score, immunohistochemistry) had small sample sizes (n = 6), and details of randomization and blinded implementation were not specified.

5. The transfection efficiency of miR-145a-5p mimic/inhibitor needs to be further verified by RNA FISH or qPCR to ensure the validity of functional experiments.

6. Some marked significant differences (e.g., * p < 0.05) do not specify the statistical test (e. g. ANOVA or T-test).

7. Mechanical stimulation does not refer to unstimulated controls (e.g., 0% stretch) and may affect the interpretation of results.

8. Some statements are not clear enough (for example, “Physiological mechanical stimulation promotes but excessive mechanical stimulation inhibits its expression” can be simplified as “Moderate stretching promotes expression, excessive stretching inhibits expression”). Please polish the language to make it more readable.

9. Specific operational details of anesthesia and euthanasia (e. g. dosage, qualification of the person performing the procedure) need to be clearly described in the method to meet animal welfare standards.

10. Limitations of the discussion section:
1) The differences between this study and other miRNAs (such as miR-145, miR-21) in OA are not fully compared, and the uniqueness of miR-145a-5p needs to be discussed.
2) The discussion of “Mechanical stress regulation of miR-145a-5p” only stays at the phenomenological level, lacks association analysis with known mechanosensitive molecules (such as PIEZO1, Yap/Taz), and the depth of the mechanism is insufficient.
3) The limitations section mentions “Apoptosis/autophagy not studied” but does not specify future research directions, suggesting additions to enhance research coherence.

11. Unify terminology spelling (e.g., “inûammation” should be “inflammation”) and fix grammatical errors throughout the text (e.g., “rescues destabilization of the medial meniscus-induced OA progression” could be optimized as ”alleviates DMM-induced OA progression “).

Add the number of experimental replications (e.g., “three independent experiments” refers specifically to technical or biological replications), and clarify the basis for the choice of statistical methods (e.g., why ANOVA was used instead of the Kruskal-Wallis test).

---

## Round 0.2 · accepted · Accept

Dear Dr. Du,

The revised version of your manuscript has been re-reviewed by the original reviewers. In light of both the reviewers' recommendations, your manuscript is accepted for publication.

Thanks,
Anoop Rawat

Reviewer 1 ·

Basic reporting

All my previous comments are addressed.

Experimental design

All my previous comments are addressed.

Validity of the findings

All my previous comments are addressed.

Additional comments

All my previous comments are addressed.

·

Basic reporting

The author responded to each of the questions I asked and was generally satisfied

Experimental design

The author responded to each of the questions I asked and was generally satisfied

Validity of the findings

The author responded to each of the questions I asked and was generally satisfied

Additional comments

The author responded to each of the questions I asked and was generally satisfied